# Theory of MBE Growth of Nanowires on Adsorbing Substrates: The Role of the Shadowing Effect on the Diffusion Transport

**DOI:** 10.3390/nano12071064

**Published:** 2022-03-24

**Authors:** Vladimir G. Dubrovskii

**Affiliations:** Faculty of Physics, St. Petersburg State University, Universitetskaya Embankment 13B, 199034 St. Petersburg, Russia; dubrovskii@mail.ioffe.ru

**Keywords:** III-V nanowires, molecular beam epitaxy, surface diffusion, shadowing effect, growth modeling

## Abstract

A new model for nanowire growth by molecular beam epitaxy is proposed which extends the earlier approaches treating an isolated nanowire to the case of ensembles of nanowires. I consider an adsorbing substrate on which the arriving growth species (group III adatoms for III-V nanowires) may diffuse to the nanowire base and subsequently to the top without desorption. Analytical solution for the nanowire length evolution at a constant radius shows that the shadowing of the substrate surface is efficient and affects the growth kinetics from the very beginning of growth in dense enough ensembles of nanowires. The model fits quite well the kinetic data on different Au-catalyzed and self-catalyzed III-V nanowires. This approach should work equally well for vapor-liquid-solid and catalyst-free nanowires grown by molecular beam epitaxy and related deposition techniques on unpatterned or masked substrates.

## 1. Introduction

Nanowires (NWs) and in particular III-V semiconductor NWs are promising for fundamental studies and applications in nanoelectronics and nanophotonics, often integrated with silicon platforms [1,2,3,4,5,6,7,8]. These NWs are synthesized by different epitaxy techniques using the vapor–liquid–solid (VLS) growth catalyzed by Au [9] or group III [10] droplets, or catalyst-free selective area growth (SAG) [11]. In some cases, the VLS and SAG III-V NWs can even co-exist in one sample depending on the NW radius and growth conditions [12]. When a liquid droplet solidifies at a low temperature, the VLS growth is transitioned to the vapor–solid–solid mode [13]. The achieved level of control over the NW dimensions, morphology, and crystal phase has been closely related to the NW growth modeling (see Ref. [14] for a review). Most works on this topic [15,16,17,18,19,20,21,22,23,24,25,26,27,28,29], with only a few exceptions [30,31,32,33,34,35], addressed the growth kinetics of individual NW without accounting for the shadowing effect [30], which is very important in molecular beam epitaxy (MBE) and related directional deposition techniques where the material beams are inclined at an angle with respect to the substrate normal.

Recently, I presented an analytic model for NW growth kinetics in the MBE technique, which addresses the shadowing effect on a reflecting substrate from which the growth species (group III atoms in the case of III-V NWs) may re-emit and land on the NW sidewalls [35]. The reflecting surface should be a good model for the substrate masked with an oxide layer for NWs grown in regular arrays of patterned pinholes at high enough temperatures [31,32,33]. On the other hand, low temperature MBE growth on patterned Si substrates may be driven by surface diffusion of group III adatoms from the substrate surface to the NW base and subsequently to the top. This type of material exchange between the substrate surface and NWs definitely occurs when NWs are grown on unpatterned substrates [16,18,19,20,21,22,24,25]. Consequently, here I develop a model for the NW growth kinetics in MBE on adsorbing substrates, the case where the diffusion transport is the main mechanism governing the NW growth. Desorption of group III atoms is considered negligible. In this case, any deposited group III atom incorporates either into NWs or parasitic structures growing between the NWs.

The study is restricted to NW elongation at a constant radius, without any radial growth. The influence of the radial growth can be ignored for Au-catalyzed III-V NWs whose length is shorter than the diffusion length of group III adatoms on the NW sidewalls, limited by surface incorporation [14,16,18,19,20,21,23]. The radial growth of self-catalyzed III-V NWs often occurs even for short NWs [14], which is explained by the droplet extension at the NW top and subsequent step flow on the NW sidewalls directed from the NW top to its base [33]. In any case, the radial growth requires a separate treatment and will be considered elsewhere. Analytical expressions for the NW length versus time and radius in different stages are presented, which correct the earlier results [14,15,16,17,18,19,20,21,22,23,24,25,26,27,36] for the shadowing effect in MBE growth and more accurate description of the diffusion transport in the absence of desorption. In particular, the classical problems of the exponential versus linear time evolution of the NW length and the length–radius dependences in the diffusion-induced growth of NWs [14,16,18,19,20,21,22,23,25,31,36] are reconsidered in the directional deposition techniques such as MBE, and in the absence of desorption of group III atoms from the substrate surface, NW sidewalls and droplets. The latter assumption requires that the growth temperature is not too high, for example lower than 620 °C for GaAs or GaP NWs [16,20,33,35]. When desorption of group III atoms becomes effective, the total material balance is broken. Furthermore, the diffusion length of group III adatoms on the NW sidewalls may become limited by desorption rather than radial growth. It is shown that shadowing of the substrate surface is very important from the very beginning of growth in dense ensembles of NWs. The model fits the kinetic data on VLS GaAs and InP NWs quite well. The shadowing effect is introduced as in Ref. [35], but in the case of adsorbing rather than reflecting substrates. Overall, this approach should be useful for the growth modeling and morphological control over both VLS and SAG NWs in a wide range of material systems.

## 2. Model

The model geometry is illustrated in Figure 1. I consider an ensemble of identical NWs having cylindrical shape with length L and radius R. The contact angle of apical droplets resting on tops of VLS NWs equals β. Catalyst-free SA NWs have no droplets on their tops. Separation between the NWs is determined by the surface area per NW $cP^{2}$, where P is the pitch of regular array and c is the shape constant (for example, c=1 for square array and $c=\sqrt{3}/2$ for hexagonal array of patterned pinholes). In the case of MBE grown on unpatterned substrates, the surface density of irregular NWs N is related to pitch as $N=1/\left(cP^{2}\right)$. For a rotating substrate, the model is relevant for both regular arrays of NWs grown in patterned pinholes, or irregular ensembles of NWs grown on unpatterned substrates. The beam angle of a group III flux with respect to the substrate normal, determined by the geometry of MBE system, equals α. The two-dimensional (2D) equivalent growth rate v is determined by the flux of an element which limits the growth (group III atomic flux for Au-catalyzed VLS NWs [16,18,19,20,21,22,23,24] or catalyst-free SA NWs [9]). For self-catalyzed III-V NWs, the axial growth rate is determined by the incoming flux of group V atoms and their desorption from the droplet [10,37]. However, the total balance of group III atoms is still relevant [33,35,38,39]. I will assume the absence of desorption of group III atoms from the substrate surface, NW sidewalls and catalyst droplets resting on the tops of VLS NWs, as discussed above. In the case of an adsorbing substrate, the diffusive growth species are collected by the droplet, NW sidewalls and also from a feeding ring of width $\lambda_{s}$ on the substrate surface [18,21,23,26,29,38]. This $\lambda_{s}$ can be called the diffusion length of group III adatoms, which describe the temperature-dependent surface mobility of these adatoms on a given substrate surface. Outside these rings surrounding the NWs, the deposited material agglomerates in the form of parasitic islands (on masked surfaces [24,38]) or a quasi-2D layer (on unpatterned surfaces [16,18,23]). The NW shape is assumed cylindrical, with the length L above the substrate surface and uniform radius R from base to top. These assumptions are normal in the growth modeling of NWs on adsorbing substrates [14]. Of course, the masked substrates can be adsorbing only at low enough temperatures—otherwise, the material exchange between the substrate and NWs occurs via reflection of group III atoms from the masked surface [31,32,33]. Additionally, I consider the ideal ensemble of NWs having identical lengths and radii, as in Ref. [35].

The main goal of this work is theoretical analysis of the total material current into the NW (including the droplet for VLS NWs), F, similarly to Refs. [33,35] for reflecting substrates. The total current equals the derivative of the NW volume plus the droplet volume with respect to time
(1)$$F=\frac{d}{dt}\left[\pi R^{2}L+\frac{\pi R^{3}}{3}f\left(\beta \right)\right].$$

Here, f(β)=(1−cosβ)(2+cosβ)/[(1+cosβ)sinβ] is the geometrical function of the droplet contact angle β [14]. For catalyst-free SAG NWs, f(β)=0 and only the first term remains in the right side of Equation (1). The total current into the NW is the sum of two currents
(2)$$F=F_{dir}+F_{surf}.$$

Here, $F_{dir}$ is the direct current originating from group III atoms impinging onto the droplet (for VLS NWs) or top NW facet (for SAG NWs), and NW sidewalls. $F_{surf}$ is the current originating from the diffusion flux from the substrate surface to the NW. The maximum current into the NW is given by the product of the 2D equivalent growth rate and the surface area per NW [33,34,35]
(3)$$F_{max}=vcP^{2}.$$

In the directional deposition methods such as MBE, this maximum current is reached when the substrate is fully shadowed by NWs so that no group III atoms can arrive onto the substrate surface [33,35].

The total balance of group III atoms in the absence of their desorption yields
(4)$$v-\frac{F_{dir}}{cP^{2}}=v_{surf}\left(1-\frac{\pi R^{2}}{cP^{2}}\right).$$

Both sides of this expression give the total material flux landing on the substrate surface. $F_{dir}/\left(cP^{2}\right)$ is the fraction of the total flux v remaining in NWs due to direct impingement and $v_{surf}$ as the flux coming onto the substrate surface free of NWs, described by the factor $1-\pi R^{2}/cP^{2}$ in Equation (4). The direct current is given by $F_{dir}=vS_{nw}$, where $S_{nw}$ is the effective collection area of the NW sidewalls and droplet. This representation is valid for $S_{nw}\leq cP^{2}$, while for $S_{nw}>cP^{2}$ the direct current equals its maximum value $cP^{2}$, similarly to Ref. [35]. For the diffusion current from the substrate, I use $F_{surf}=v_{surf}S_{coll}$, where $S_{coll}$ is the effective collection area for the diffusive species on the substrate surface. Using Equation (1) and expressing $v_{surf}$ through v from Equation (4), one obtains
(5)$$F=v\left[S_{nw}+\frac{\left(cP^{2}-S_{nw}\right)S_{coll}}{cP^{2}-\pi R^{2}}\right],\;S_{nw}\leq cP^{2}\;F=vcP^{2},\;S_{nw}>cP^{2}.$$

Without specifying the form of $S_{nw}$ and $S_{coll}$, Equation (5) correctly reproduces the two important limiting cases. When $S_{coll}=0$, one simply has $F=vS_{nw}$, meaning no diffusion from the substrate surface to NW (a fraction of material collected by the NW sidewalls may, however, diffuse to the substrate) [21,23,25]. On the other hand, when $S_{coll}=cP^{2}-\pi R^{2}$, that is, group III atoms are collected from the whole substrate surface free of NWs, one has $F=vcP^{2}$. In this case, the total current into the NW equals its maximum at any time, because no group III atoms are lost for parasitic growth or desorption.

To obtain $S_{nw}$ and $S_{coll}$, we consider the usual set of the two steady-state diffusion equations for the group III adatom concentrations on the NW sidewalls ($n_{f}$) and the substrate surface ($n_{s}$) [25], which simplify to
(6)$$\frac{d^{2}n_{f}}{dz^{2}}=-\frac{Jsin\alpha }{\pi D_{f}},\;\frac{d^{2}n_{s}}{dr^{2}}+\frac{1}{r}\frac{dn_{s}}{dr}=-\frac{Jcos\alpha }{D_{s}}$$
in the absence of desorption or surface growth. The absence of radial growth on the NW sidewalls implies that the NW radius R=const. Here, z is the vertical coordinate along the NW axis and r is the 2D radius-vector, z=0 corresponds to the substrate surface and r=0 to the NW center. $D_{f}$ and $D_{s}$ are the diffusion coefficients of adatoms on the NW sidewalls and substrate surface. Atomic flux J per unit surface area is related to the 2D equivalent growth rate as v=ΩJcosα. General solutions to Equation (6) are given by
(7)$$n_{f}=-\frac{Jsin\alpha }{2\pi D_{f}}z^{2}+a_{1}z+a_{2},\;n_{s}=-\frac{Jcos\alpha }{4D_{s}}r^{2}+b_{1}lnr+b_{2},$$
with the four coefficients determined by the boundary conditions.

The diffusion-induced contribution to the total current is given by [25]
(8)$$F_{diff}=-2\pi \Omega D_{f}R{\frac{dn_{f}}{dz}|}_{z=L}.$$

Using the continuity equation for the diffusion flux at the NW base [19,20,25],
(9)$$D_{s}{\frac{dn_{s}}{dr}|}_{r=R}=-D_{f}{\frac{dn_{f}}{dz}|}_{z=0},$$
one obtains
(10)$$F_{diff}=2RLvtan\alpha -\pi R^{2}v+2\pi \Omega D_{s}b_{1}.$$

Clearly, the first term equals the total area of the NW sidewalls exposed to the flux. The diffusion current depends on the coefficient $b_{1}$ in Equation (7) for $n_{s}$, which is obtained using the second boundary condition at the periphery of the collection ring
(11)$$2\pi \Omega D_{s}\left(R+\lambda_{s}\right){\frac{dn_{s}}{dr}|}_{r=R+\lambda_{s}}=-\left(1-\xi \right)2RLvtan\alpha -\left(1-\varphi \right)\pi \left[{\left(R+\lambda_{s}\right)}^{2}-R^{2}\right].$$

This boundary condition accounts for the possible reverse flux from the NW sidewalls and collection ring. The fractions 1−ξ and 1−φ of the total amount of group III atoms adsorbed by the NW sidewalls and the collection ring diffuse across the boundary of the collection ring at $r=R+\lambda_{s}$ and contributing to the parasitic growth on the substrate surface [20,21,23,25]. Using Equation (11) for determination of $b_{1}$ and inserting the result into Equation (10), the diffusion current is obtained in the form
(12)$$F_{diff}=\xi 2RLvtan\alpha +\varphi \pi \left[{\left(R+\lambda_{s}\right)}^{2}-R^{2}\right]v.$$

According to this expression, the fraction ξ of the total amount of group III atoms collected by the NW sidewalls diffuse to the NW top, while the fraction φ of the total amount of group III atoms impinging onto the collection ring on the substrate surface diffuse to the NW base and subsequently to the top, as in Ref. [21] where a similar expression was introduced empirically. The impingement flux onto the droplet surface equals $\left(\chi /cos\alpha \right)\pi R^{2}$, where χ is the geometrical factor of MBE growth which is determined by the droplet contact angle β and the beam angle *α* [40]. Adding the direct impingement term to the diffusion current given by Equation (12), the $S_{nw}$ and $S_{coll}$ are obtained in the form
(13)$$S_{nw}=\frac{\chi }{cos\alpha }\pi R^{2}+\xi 2RLtan\alpha ,\;S_{coll}=\varphi \pi \left[{\left(R+\lambda_{s}\right)}^{2}-R^{2}\right].$$

Using these expressions in Equation (5), one arrives at the final result of the form
(14)$$\frac{F}{v}=\frac{\varphi \pi \left[{\left(R+\lambda_{s}\right)}^{2}-R^{2}\right]}{cP^{2}-\pi R^{2}}cP^{2}+\frac{cP^{2}-\pi R^{2}-\varphi \pi \left[{\left(R+\lambda_{s}\right)}^{2}-R^{2}\right]}{cP^{2}-\pi R^{2}}\left[\frac{\chi }{cos\alpha }\pi R^{2}+\xi 2RLtan\alpha \right],\;\frac{\chi }{cos\alpha }\pi R^{2}+\xi 2RLtan\alpha \leq cP^{2};\;\frac{F}{v}=cP^{2},\;\frac{\chi }{cos\alpha }\pi R^{2}+\xi 2RLtan\alpha >cP^{2}.$$

This result for the total current contains the three main geometrical factors—the droplet surface, the surface of NW sidewalls, and the surface of the collection ring on the substrate, similarly to Ref. [26]. However, it generalizes the earlier results for the diffusion transport of material into the NW for the two factors—(i) the shadowing effect on the substrate surface and (ii) the maximum current which is simply determined by the surface area per NW after the diffusion exchange with the substrate is blocked. Therefore, the axial NW growth rate gradually increases with increase in the portion of the shaded area on the substrate surface and becomes a pitch-dependent constant $cP^{2}$ after the full shadowing of the substrate.

As mentioned above, the NW radius *R* should be kept constant during growth in the absence of surface incorporation on the NW sidewalls. This is the usual assumption in the NW growth modeling and holds for Au-catalyzed VLS III-V NWs whose length is shorter than the diffusion length of group III atoms on the NW sidewalls (limited by the radial growth) [15,16,17,18,19,20,21,22,23,24,25,26,27,29,31,41]. Depending on the V/III flux ratio and temperature, the diffusion length of Ga adatoms on the sidewalls of GaAs NWs was estimated at around 1500 nm in Ref. [21], 1800 nm in Ref. [38], and more than 3000 nm in Ref. [16]. The diffusion length of In adatoms on the sidewalls of InP_1−x_As_x_ NWs was estimated at more than 2000 nm in Ref. [23]. Self-catalyzed VLS III-V NWs [33,37,38,39,42,43] often grow radially from the very beginning. This feature is due to the fact that a catalyst droplet serves as a non-stationary reservoir of group III atoms and can either swell or shrink depending on the V/III flux ratio and surface diffusivity of group III adatoms [42,43]. The droplet volume can change by changing either the radius of the NW top or contact angle, depending on the surface energetics [44]. In the former case, the NW radius enlarges by step flow starting from the NW top [33]. In the latter case, the NW radius remains constant, while the change in the droplet volume becomes negligible compared to the change in the NW volume for high enough aspect ratios L/R. Catalyst-free SAE III-V NWs also extend radially from the very beginning of growth [11]. Below we will study the simplified case of R=const and β=const, leaving the refinement for further studies. From Equation (1), one has $F=\pi R^{2}dL/dt$, and Equation (14) gives
(15)$$\frac{dL}{dH}=\frac{\varphi \pi \left[{\left(R+\lambda_{s}\right)}^{2}-R^{2}\right]}{cP^{2}-\pi R^{2}}\frac{cP^{2}}{\pi R^{2}}+\frac{cP^{2}-\pi R^{2}-\varphi \pi \left[{\left(R+\lambda_{s}\right)}^{2}-R^{2}\right]}{cP^{2}-\pi R^{2}}\left[\frac{\chi }{cos\alpha }+\xi \frac{2tan\alpha }{\pi }\frac{L}{R}\right],$$
with H=vt as the effective deposition thickness.

Integrating this with the initial condition, one obtains
(16)$$L\left(H\right)=\left\{\frac{\chi \pi R}{2\xi sin\alpha }+\frac{\varphi \pi \left[{\left(R+\lambda_{s}\right)}^{2}-R^{2}\right]}{cP^{2}-\pi R^{2}-\varphi \pi \left[{\left(R+\lambda_{s}\right)}^{2}-R^{2}\right]}\frac{cP^{2}}{\pi R^{2}}\frac{\pi Rcotan\alpha }{2\xi }\right\}\;\times \left\{exp\left[\frac{cP^{2}-\pi R^{2}-\varphi \pi \left[{\left(R+\lambda_{s}\right)}^{2}-R^{2}\right]}{cP^{2}-\pi R^{2}}\frac{2\xi tan\alpha }{\pi }\frac{H}{R}\right]-1\right\},\;0\leq L\leq L_{*}.$$

This solution is valid for NW lengths shorter than the critical shadowing length
(17)$$L_{*}=\frac{cotan\alpha }{2\xi R}cP^{2}-\frac{\chi \pi R}{2\xi sin\alpha }$$
corresponding to $S_{nw}=cP^{2}$. Longer NWs evolve according to
(18)$$L=L_{*}+\frac{cP^{2}\left(H-H_{*}\right)}{\pi R^{2}},\;L>L_{*},$$
where $H_{*}$ is the deposition thickness at $L=L_{*}$. Equations (16)–(18) give the solution to the classical problem of the elongation law for a NW at a constant radius and droplet volume [34] in the case of MBE growth of an ensemble of NWs. The shadowing effect on the collection of group III adatoms on the substrate surface manifests in the pitch-dependent terms describing surface diffusion from the surface to NW sidewalls. This approach is different from the earlier works (see, for example, Ref. [30]), where the shadowing started from a certain NW length corresponding to blocking of group III flux impinging the NW sidewalls.

## 3. Results and Discussion

In the limiting case P→∞ Equation (16) is reduced to
(19)$$L\left(H\right)=\left[\frac{\chi \pi R}{2\xi sin\alpha }+\frac{\pi Rcotan\alpha }{2\xi }\varphi \left(\frac{2\lambda_{s}}{R}+\frac{\lambda_{s}^{2}}{R^{2}}\right)\right]\left[exp\left(\frac{2\xi tan\alpha }{\pi }\frac{H}{R}\right)-1\right],\;0\leq L\leq L_{*},$$
which is the result of Ref. [21] for isolated NW, modified for an arbitrary droplet contact angle. At $\lambda_{s}\to 0$, Equation (16) simplifies to
(20)$$L\left(H\right)=\frac{\chi \pi R}{2\xi sin\alpha }\left[exp\left(\frac{2\xi tan\alpha }{\pi }\frac{H}{R}\right)-1\right],\;0\leq L\leq L_{*},$$
showing that the NW elongates by collecting group III atoms from the droplet surface and NW sidewalls until it reaches the critical length. At φ=1, where all group III atoms impinging onto the collection ring are subsequently incorporated into the NW, Equation (16) takes the form
(21)$$L\left(H\right)=\left[\frac{\chi \pi R}{2\xi sin\alpha }+\frac{\pi {\left(R+\lambda_{s}\right)}^{2}-\pi R^{2}}{cP^{2}-\pi {\left(R+\lambda_{s}\right)}^{2}}\frac{cP^{2}}{\pi R^{2}}\frac{\pi Rcotan\alpha }{2\xi }\right]\left\{exp\left[\frac{cP^{2}-\pi {\left(R+\lambda_{s}\right)}^{2}}{cP^{2}-\pi R^{2}}\frac{2\xi tan\alpha }{\pi }\frac{H}{R}\right]-1\right\}.$$

Exponential growth of the NW length with H can be suppressed for short enough growth times corresponding to low H, or small ξ, in which case most group III atoms landing on the NW sidewalls diffuse to the substrate surface rather than to NW top [20]. At (2ξtanα/π)(H/R) ≪ 1, Equation (21) is reduced to
(22)$$L\left(H\right)=\left[\frac{\chi }{cos\alpha }\frac{cP^{2}-\pi {\left(R+\lambda_{s}\right)}^{2}}{cP^{2}-\pi R^{2}}+\frac{cP^{2}}{\pi R^{2}}\frac{\pi {\left(R+\lambda_{s}\right)}^{2}-\pi R^{2}}{cP^{2}-\pi R^{2}}\right]H,$$
the growth law which is linear in H. In the limit P→∞, this is further simplified to
(23)$$L\left(H\right)=\left[\frac{\chi }{cos\alpha }+\frac{2\lambda_{s}}{R}+\frac{\lambda_{s}^{2}}{R^{2}}\right]H.$$

According to this equation, the NW elongates due to the direct impingement onto the droplet surface or top facet, and by collecting group III atoms from the ring of width $\lambda_{s}$ on the substrate surface [25]. Depending on the ratio $\lambda_{s}/R$, such a growth may correspond to $R^{-1}$ (at $\lambda_{s}/R\ll 1$) or $R^{-2}$ (at $\lambda_{s}/R\gg 1$) radius dependence of the NW length [17,18,25].

At $\lambda_{s}\to 0$, which often occurs for III-V NWs growing on an unpatterned substrate where a parasitic quasi-2D layer forms everywhere between the NWs [23], the total material balance in the absence of desorption of group III atoms yields $H=\pi R^{2}/\left(cP^{2}\right)+[1-\pi R^{2}/\left(cP^{2}\right)]H_{2D}$. Here, $H_{2D}$ is the mean thickness of the parasitic layer which can be directly measured [16,21,23]. The NW length above the parasitic layer equals $L_{nw}=L-H_{2D}$. Using the total balance of group III atoms, one obtains
(24)$$L_{nw}\left(H\right)=\frac{L\left(H\right)-H}{1-\pi R^{2}/cP^{2}},$$
where L(H) is calculated using Equations (16) and (18) or one of the approximations given above.

Figure 2a shows the NW length as a function of the deposition thickness obtained from Equations (16)–(18) for MBE-grown VLS NWs at a beam angle of 35°, droplet contact angle of 125° (corresponding to $\chi =1/sin^{2}\beta =1.491$ [40]), for a square array (c=1) of patterned pinholes with a pitch P of 300 nm at a width of collection ring of 50 nm for different NW radii from 15 to 100 nm. The NW length strongly increases for smaller NW radii for a given deposition thickness or time, as usual in the diffusion-induced growth [14,15,16,18,21,25,36]. However, the length–radius correlation in the general case is more complex compared to the typical dependences discussed earlier (such as $R^{-1}$ or $R^{-2}$ correlation) and depends on the pre-history of NW growth. Evolution of the NW length with the deposition thickness or time is exponential at the beginning, converging to linear dependence after the full shadowing of the substrate surface. According to Equation (17), the critical length for this transition is larger for a smaller NW radius. Consequently, the L(H) curves change from exponential for the smallest radius of 15 nm to an almost linear for the largest radius of 100 nm.

Figure 2b shows the NW length versus the deposition thickness for a fixed NW radius of 25 nm and different coefficients ξ from zero (corresponding to zero adatom collection by the NW sidewalls) to unity (the maximum collection). Other parameters are the same as in Figure 2a. The linear curve at ξ=0, given by Equation (22), is transitioned to non-linear curves for larger ξ, which start from exponential and then converge to linear at the critical lengths. As expected, a more efficient material collection at larger ξ yields longer NWs for a given deposition thickness or time. Figure 2c shows the L(H) dependence at a given NW radius of 25 nm and ξ=1 for three different pitches of 200 nm, 300 nm and 400 nm. Since the critical length given by Equation (17) decreases for smaller pitches, the L(H) curves become more linear toward smaller P. At a given H, the NW length increases with the pitch for two reasons. First, the transition from exponential to linear growth occurs later due to a larger critical length. Second, the shadowing effect on the collection rings on the substrate surface is weakened for larger separation between the NWs. Figure 2d shows how the NW increases with increasing $\lambda_{s}$ at a fixed NW radius of 25 nm and a pitch of 300 nm. It is interesting to note that the L(H) curves contain longer exponential sections for shorter $\lambda_{s}$. At a maximum $\lambda_{s}$ of 144 nm, the L(H) curve is linear at any H. This corresponds to the adatom collection from the whole substrate surface ($\pi {\left(R+\lambda_{s}\right)}^{2}=cP^{2}$), where the evolution of the NW length is given by Equation (18) with $L_{*}=0$. As for the dependence of the NW length on the beam angle α, the length generally increases with α due to a more efficient material collection on the NW sidewalls. On the other hand, full shadowing of the substrate surface occurs earlier for larger α, as given by Equation (17) and discussed in detail in Ref. [35].

## 4. Theory and Experiment

We first consider the length–radius dependencies of III-V NWs grown by MBE on unpatterned substrates. Although we assumed a narrow radius distribution within the ensemble of NWs for the shadowing effect, Equations (19) and (20) describe the growth of isolated NWs and hence can be used for modeling the length–radius curves for NWs with a low surface density or short lengths. Furthermore, Equation (20) does not account for any material exchange with the substrate surface at $\lambda_{s}\to 0$. The absence of surface diffusion is supported by the data of Ref. [23] for III-V NW growth on unpatterned substrates, where surface adatoms are rapidly captured by quasi-2D parasitic layer growing between the NWs. Au-catalyzed GaAs NWs of Ref. [16] were grown by solid source MBE on an unpatterned GaAs(111)B substrate, where Au droplets were obtained by thermal annealing of Au film. The growth temperature was 585 °C and the deposition thickness H amounted to 270 nm, at *α* = 30°. The surface density of NWs *N* was about 2 × 10^8^ cm^−2^, which corresponds to *P* = 707 nm at *c* = 1. Figure 3a shows the measured length–diameter correlation and its fit by Equations (20) and (24) at *β* = 120° ($\chi =1/sin^{2}\beta =1.333$) and ξ=1. From fitting the data, the sidewall collection of Ga atoms in this case occurs with 100% probability. Consequently, the magnifying factor of NW growth, which equals the ratio of the NW length over the deposition thickness, L/H reaches 17.4 for the thinnest NWs of ~30 nm radius. The dashed in the figure shows the best fit obtained from the linear growth law $L=(\chi /cos\alpha +\lambda_{f}/R)H$, with $\lambda_{f}$ as the diffusion length of Ga adatoms on the NW sidewalls [25], at $\lambda_{f}=$ 433 nm. This fit definitely does not work. The dashed–dotted line shows the fit obtained from the linear growth law with the $R^{-2}$ diffusion term, $L=(\chi /cos\alpha +\Lambda_{s}^{2}/R^{2})H$ [25], at $\Lambda_{s}=$ 96 nm. This fit is much better and very close to the exponential curve. However, such efficient material collection from the unpatterned substrate surface is unlikely [23].

Figure 3b shows the data on Au-catalyzed InP NWs of Ref. [41]. These NWs were grown by gas-source MBE on an unpatterned InP(111)B substrate at 420 °C. The effective 2D thickness of InP was 250 nm, at *α* = 35°. Figure 2b shows the excellent fit to the data obtained from Equations (20) and (24) at *β* = 125° and ξ=0.446. In this case, the sidewall collection of In atoms is less efficient than for Ga atoms in the previous example. Consequently, the magnifying factor L/H reaches only 8.8 for the narrowest NWs of ~13 nm radius, although these NWs are much thinner than in Figure 3a.

Growth kinetics of NWs can be understood deeper through in situ [38] or ex situ [21,31,33] studies of the morphological evolution with the growth time or deposition thickness. GaAs NWs of Ref. [21] were grown by gas-source MBE at 600 °C, on unpatterned GaAs(111)B substrates, where the Au droplets were obtained by thermal annealing of Au film. The Ga beam angle was 35°. The growth times were varied, corresponding to the range of deposition thickness shown in Figure 4a. These data points correspond to untapered NWs without radial growth (which started for NWs longer than 1500 nm) and present the average values for the NW length. The average diameter of the NWs in this growth stage was 44 nm. The measured L(H) curve is markedly non-linear, and well-fitted by Equation (21) at φ=1, R= 22 nm, $\lambda_{s}=$ 6 nm, P=300 nm, and ξ=0.10. The pitch corresponds to the data on the NW density, while $\lambda_{s}$ and ξ are selected for the best fit of the curve shown in the figure. The sidewall collection of Ga adatoms leads to the exponential shape of the curve, although only 10% of the total Ga flux intercepted by NW contributes into its elongation. The magnifying factor L/H reaches only 3.4 at H= 500 nm.

Figure 4b shows the data on the length evolution of a single self-catalyzed GaAs NW, obtained by in situ x-ray diffraction technique in Ref. [38]. This GaAs NW was grown by MBE in patterned arrays of pinholes in SiO_x_/Si(111) with Ga pre-deposition, at a temperature of 610 °C. The Ga bean angle was 30°. The NWs were grown concomitantly with parasitic GaAs nano-islands that formed unintentionally on the oxide mask surface. The NWs and islands started to form after an incubation time of 22.5 min. A constant NW radius R of 14 nm was maintained in the growth stage shown in the figure, with the onset of radial growth at a length of 1800 nm. It can be seen that the measured L(H) curve is linear, in sharp contrast with Figure 4a. The NW density was extremely low, corresponding to a pitch of 5000 nm. Therefore, linear evolution of the NW length with time can be explained by the absence of sidewall collection of Ga atoms, which corresponds to (2ξtanα/π)(H/R)≪1. In this case, the growth law of isolated NW is given by Equation (23), with the best fit obtained at $\lambda_{s}=$ 88.5 nm. With this large collection length on the surface of oxide layer, the magnifying factor L/H reaches 55.4, which is much higher than in the previous case.

SAG of Au-catalyzed InP NWs of Ref. [31] was performed by chemical beam epitaxy in hexagonal arrays of patterned holes in SiO_2_ on InP(111)B substrates at 420 °C. The growth time was 15 min, corresponding to H= 58 nm thick for the data shown in Figure 5. The In beam angle *α* was 45°. InAs markers were used to measure the growth kinetics of a thin InP NW with an approximately uniform radius *R* = 12 nm form base to top. The blue line in the figure shows the fit obtained using the model of reflecting substrate of Ref. [35], assuming the Lambertian re-emission of In atoms from the oxide mask. The green line shows the best fit with Equations (21) and (18), obtained at *P* = 175 nm, *ξ* = 1, and $\lambda_{s}=$ 15 nm. The shadowing length $L_{*}$ for these parameters equals 1250 nm. The fit is close to the one obtained for reflecting substrate, although the correspondence is better in the latter case. One can thus conclude that the growth mechanism of these InP NWs is most likely associated with In re-emission from the mask. A similar conclusion was drawn in Ref. [33] for Ga-catalyzed GaP NWs grown by MBE at 600 °C in regular arrays of patterned holes in SiO_2_ mask layer on Si(111) ($cP^{2}$ = 216,506 nm^2^), with *α* = 32.5°. magenta curve in the figure shows the minimum NW length at ξ=1 and $\lambda_{s}=$ 0, where In adatoms are collected only from the NW sidewalls. As expected, this length is much smaller compared to that obtained from the data, because the re-emitted flux of In adatoms is not included. On the other hand, if all In atoms were collected from the mask surface, the linear evolution of the NW length would largely exceed the measured length. This is demonstrated by the red curve in the figure.

## 5. Conclusions

In conclusion, an analytic model for NW growth has been developed which accounts for the shadowing effect on the diffusion transport of group III adatoms from the substrate surface to NWs in MBE and related directional deposition methods. It has been shown that MBE growth of dense ensembles of NWs is influenced by the shadowing effect from the very beginning, which significantly modifies the growth models considered earlier. The maximum axial growth rate of NWs is achieved in the linear growth regime where all group atoms impinging on the substrate surface are subsequently collected by NWs. This growth rate is larger than for NWs growing on a reflecting substrate. Exponential time evolution of the NW length is observed when a fraction of group III atoms remains in the parasitic layer. The model fits quite well the data on the length evolution of different III-V NWs growing at a time independent radius on different substrates and with different catalysts, as well as the length–radius dependencies at a given time. We now plan to study the radial growth of NWs using a similar approach, because ignoring the surface incorporation on the NW sidewalls cannot be justified in the entire range of NW lengths, particularly for self-catalyzed III-V NWs. Another refinement should regard the size variation of NWs in terms of both lengths and radii. Overall, the developed approach is quite general and should work equally well for both VLS and SAG NWs grown on unpatterned or masked surfaces, in a wide range of material–substrate combinations.

## Figures and Tables

**Figure 1 nanomaterials-12-01064-f001:**
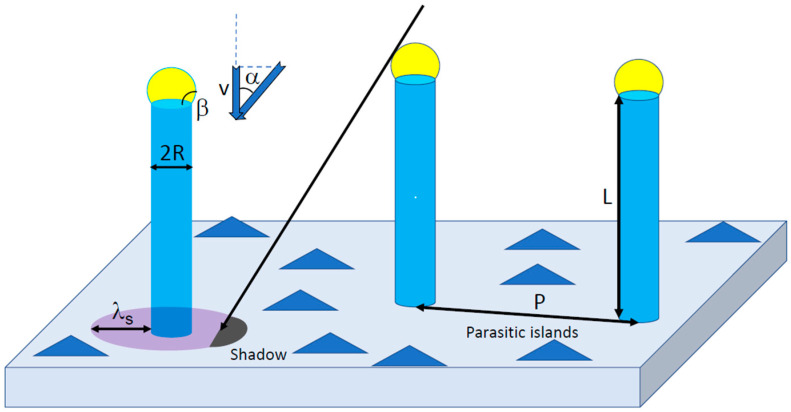
Ensemble of identical VLS NWs with length L and radius R. The contact angle of the catalyst droplets on the NW tops equals β. The molecular beam of an element which limits the NW growth is inclined at an angle α with respect to the substrate normal; v is the 2D equivalent growth rate. The NWs are fed from the collection rings of width $\lambda_{s}$ on the substrate surface around each NW. Distance between the NWs (pitch) equals P. Parasitic growth of islands or the quasi-2D layer occurs between the NWs outside the collection rings. The shadowing effect originates from the neighboring NWs which block a fraction of the collection rings.

**Figure 2 nanomaterials-12-01064-f002:**
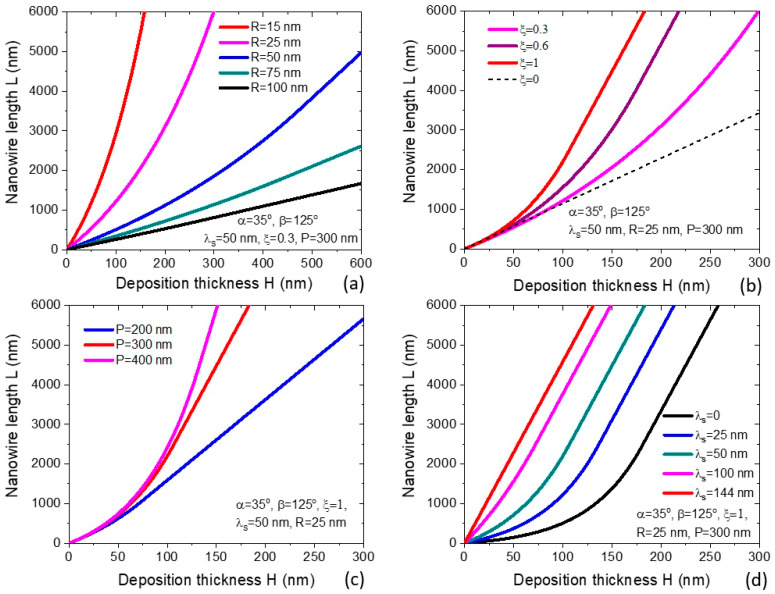
NW length versus deposition thickness obtained from Equations (16)–(18) for VLS NWs with α = 35°, β = 125°, c = 1, and φ=1 at (**a**) a fixed ξ of 0.3, $\lambda_{s}$ = 50 nm, and P = 300 nm for different NW radii shown in the caption; (**b**) a fixed $\lambda_{s}$ of 50 nm, P = 300 nm, and R = 25 nm for different coefficients ξ shown in the legend (the dashed straight line corresponds to the limiting case of ξ→0); (**c**) a fixed ξ of 1, $\lambda_{s}$ = 50 nm, and R = 25 nm for different pitches shown in the legend; and (**d**) a fixed ξ of 1, R = 25 nm, and P = 300 nm for different widths of the collection ring $\lambda_{s}$ shown in the legend.

**Figure 3 nanomaterials-12-01064-f003:**
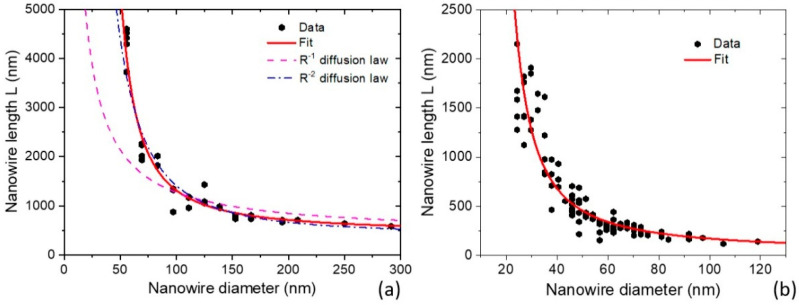
(**a**) Length–diameter dependence for Au-catalyzed GaAs NWs grown by MBE on unpatterned GaAs(111)B substrates [16]. The red line is the fit by the exponential model given by Equation (20) with ξ=1. Dashed and dash–dotted lines show the fits obtained from the linear growth laws with $R^{-1}$ and $R^{-2}$ diffusion terms. (**b**) Length–diameter dependence for Au-catalyzed InP NWs grown by MBE on unpatterned InP(111)B substrates [41]. The line shows the fit by the exponential model given by Equation (20) with ξ=0.446.

**Figure 4 nanomaterials-12-01064-f004:**
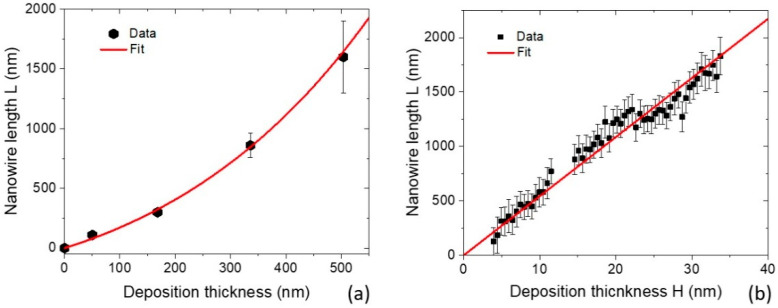
Evolution of the NW length with deposition thickness for (**a**) Au-catalyzed GaAs NWs grown by MBE on the unpatterned GaAs(111)B substrate [21], and (**b**) individual Ga-catalyzed GaAs NW grown by MBE on the patterned Si(111) substrate [38]. The length evolution is exponential in (**a**) and linear in (**b**).

**Figure 5 nanomaterials-12-01064-f005:**
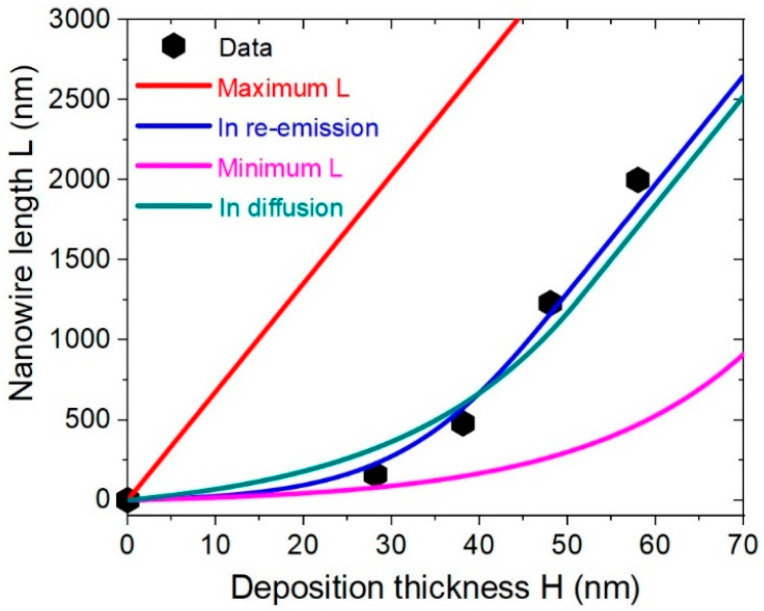
Length of an Au-catalyzed, 12 nm radius InP NW versus the deposition thickness. Symbols represent the data of Ref. [31]. The blue line is the fit obtained within the growth model of Ref. [35] for the reflecting substrate assuming the Lambertian scattering of In atoms from the oxide mask. The green line is the best fit obtained from Equations (18) and (21).

## Data Availability

Not applicable.

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
