# Peer review of "Theory of MBE Growth of Nanowires on Adsorbing Substrates: The Role of the Shadowing Effect on the Diffusion Transport"

_nanomaterials, 2022, doi:10.3390/nano12071064_

Round 1

Reviewer 1 Report

Semiconductor nanowires (NWs), especially III-V nanowires, are expected to be used in many fundamental types of research and applications in nanoelectronics and nanophotonics due to their unique optoelectronic properties. Exploring growth laws and explaining growth phenomena is significant for the efficient and controllable preparation of required nanowires. Dubrovskii created a computational model to elucidate the impacts of shadowing effect on nanowire growth dynamics which is significant in MBE and related directional deposition techniques. As note in this paper, the unique shadowing effect have tremendous influence on MBE growth of dense ensembles of NWs, ignored in most works but is considered carefully in this model. The theoretical results are consistent with the kinetic data on different Au-catalyzed and self-catalyzed III-V nanowires, which is helpful for understanding the vapor-liquid-solid and catalyst-free nanowires grown on unpatterned or masked substrates. Therefore, the computational model with the results and discussion on nanowire growth dynamics should attract attention of both theoretical and experimental scientists. Overall, the manuscript is well-written and the results and discussion seem to be reasonable. I would like to support this publication in Nanomaterials after the minor revisions. Also, questions and comments are listed as follows:

Minor comments: 1. The theoretical model is applied to the NWs elongation at a constant radius, but it is important to interpret why the influence of radius growth can be ignored in these cases, such as Au-catalyzed and self-catalyzed III-V nanowires growth. In this way, the manuscript will certainly become more appealing.

2. The author illustrates the model introducing the shadowing effect is different from other theoretical methods, however, the comparison among distinctive models should be presented to point out the impact of shadowing effect on the nanowire growth. The author needs to have a more in-depth discussion in this regard.

3. Some relevant literature are recommended to be cited. (1) Nanoscale. 2017, 9 (1), 52-55 (2) Advanced Materials. 2013, 25 (41), 5910-5915

4. There are some grammar mistakes in the manuscript, please be careful to revise it: (1) In the Page 1, line 35: “Recently, I presented an analytic model for the NW growth kinetics in MBE, which includes the shadowing effect a reflecting substrate from which the growth species (group III atoms in the case of III-V NWs) re-emit and land on the NW sidewalls.” (2) In the Page 11, line 357: “T It has been shown that MBE growth of dense ensembles of NWs…”.

5. Based on previous research, the authors introduced the shadow effect combined with the difference between the growth of a single nanowire and the growth of multiple nanowires in actual conditions. When the author introduced λs, he made a series of approximations. However, for the size of λs, the choice of material has nothing to do with the material itself. Does it have something to do with the mobility of the elements that create the nanowires on the substrate?

6. Briefly describe the relationship between the change in the shaded portion's area and the nanowire's growth rate as the nanowire grows.

7. Patterned substrate growth can guarantee the specific spacing between nanowires and the angle at which the shadow is cast, so whether the nanowire growth predicted by this modeling is reliable enough for nanowire growth on an unpatterned substrate?

8. The author believes that the shadow effect has already shown a critical role in the early stage of nanowire growth, so is the shadow effect still applicable to the development of the same nanowire on different substrate materials?

Reviewer 2 Report

    In the manuscript, a new model for nanowires growth by molecular beam epitaxy is proposed which extend the earlier approaches treating an isolated nanowire to the case of ensembles of nanowires. The model fits quite well with the experiment data on different Au-catalyzed and self-catalyzed nanowires. I have some specific comments and questions regarding the manuscripts:

  1. In this paper, the author points out that the new model extends the earlier approaches treating an isolated nanowire to the case of ensembles of nanowires. Compared with earlier approaches, I wonder the model of treating an ensemble in this study, just “consider an ensemble of identical NWs having cylindrical shape with length L and radius R”?
  2. Paragraph 3, Page 8. “Consequently, the magnifying factor of NW growth L/H reaches 17.4 for the thinnest NWs of ~30 nm radius”. I want to know what’s the magnifying factor? how to obtain the value of L/H? The author did not mention it.
  3. Fig 4 (a). I know that the author set the value of R according to the experiment data. For and P, why the author set ? Just for the best fit? Or can we get some information about them from the experimental data?
  4. Fig 5. The blue line (the earlier approach) fitted better with the experiment data. Could the author point out the cases that these two approaches work well separately?

Reviewer 3 Report

A new growth kinetic model (III-V group elements) is proposed in this paper, and the theoretical calculation of growth kinetics is carried out on the shadow effect accompanying the transport effect of adatoms in the process of growing nanowires by MBE. The relationship among the length, thickness, diameter, angle, etc. of the nanowires grown by MBE is calculated, and theoretical predictions are made. And then conducted experiments to verify the accuracy of the model proposed in the article. Overall, the model fits the experiment well, but I recommend the authors to address the following issues:

  1. In the paragraph where the model is proposed, the shadowing effect in the transmission process is proposed. I would like to ask whether the temperature (or, for example, substrate) factor should be taken into account in the model. The effect of temperature on transport should not be negligible. Or whether this model is applicable at a certain temperature.
  2. In the previous article "For self-catalyzed III-V NWs, the axial growth rate is determined by the incoming flux of group V atoms and their desorption from the droplet [8,35]. However, the total balance of group III atoms is still relevant [31, 33, 36, 37].” The growth process conditions of the elements of group III-V are different. In the derivation of the theoretical model, only the total equilibrium state of group III is used to deduce the model. This Whether the model is also applicable to MBE growth of other compounds.
  3. In the experimental verification, whether only InP catalyzed by Au is done, and in the MBE growth experiment of autocatalytic GaAs, InP is a group III element, and As is a group V element, which is not considered in the above theoretical model. For type III-V nanowires, is it correct that this experiment is close to the model predicted by theory.
  4. In the prediction of the theoretical model, the angle of ? is the prediction of the relationship between the length and diameter of the nanowire listed in the case of 35°, and there is no control variable in the graph of the theoretical prediction, such as the a and d of figure 2. In the case of changing ?s and R, the control of ? is not the same.
  5. In the experimental part, only the relationship between the length and diameter of InP and GaAs and the relationship between substrate treatment or catalysis were explored under the same conditions, and the shadow effect during MBE transmission was not explored under other conditions.
  6. Many literatures studied in this paper do not consider the shadowing effect, so what is the important effect of shadowing effect on the growth kinetics of NW? The desorption of group III atoms is ignored in this model, and does it have an impact on the final results of the model? These need to be added in the “Introduction”.
  7. During the derivation of the model establishment, the citation of the context formula should be rigorous. For example, the diffusion current depends on the coefficient of the group III adatom concentrations on the substrate surface in Equation (7) instead of Equation (78).
  8. In the “theory and experiment” section of this paper, a series of data, such as λs, are subjected to a limit approximation for the proposed model, and the conclusions obtained are fitted with the reference data. In order to make the results more credible, other situations should be briefly described.
  9. This paper mainly discusses the advantages and generality of the analytical model of NW growth kinetics in MBE including the shadowing effect of reflective substrates, but the model may have shortcomings, such as other factors that affect NW growth can be added to the model in addition to diffusion transport? For other factors not considered by the model and possible shortcomings, it is best to briefly explain at the end.

Reviewer 4 Report

The author develops an analytic model for nanowire (NW) growth by molecular beam epitaxy (MBE). This model accounts for the shadowing effect on the diffusion transport and it shows that the shadowing effect influences the growth. The model agrees well with the experimental data. Furthermore, this model is quite general and may be extended to other cases such as VLS and SAG NWs grown on unpatterned or masked substrates.

This is an interesting work and the model significantly modifies the growth models considered earlier. I recommend it for publication after the authoraddresses the following questions.

  1. In Fig. 2, the NW length is dependent on the deposition thickness H according to Eq. (16) and Eq. (18), and different cases with various parameters have been shown. I wonder what the dependence on the beam angle alpha is, because this parameter is also important and in the experiments. Can the author give some analysis on this relation?  
  2.  The y labels for Fig. 2-5 are slightly different, some are Nanowire length and the others are Nanowire length L. Maybe it is better for unified writing.
  3.  Some spelling mistakes should be addressed. For example, Page 9, Line 303, “… obtained from Eqs. Eqs. (20) and (24) …”; in Conclusion, Line 357, “ TIt has … ”
  4.  Some references may be related to this work, such as:

https://doi.org/10.1016/j.optcom.2021.127277

http://dx.doi.org/10.1088/1361-6463/ac4fd6
